# Chemical compass behaviour at microtesla magnetic fields strengthens the radical pair hypothesis of avian magnetoreception

Christian Kerpal[1,4], Sabine Richert [1,4], Jonathan G. Storey[1], Smitha Pillai[2], Paul A. Liddell[2], Devens Gust[2], Stuart R. Mackenzie [3], P.J. Hore[3] & Christiane R. Timmel [1]

The fact that many animals, including migratory birds, use the Earth's magnetic field for orientation and compass-navigation is fascinating and puzzling in equal measure. The physical origin of these phenomena has not yet been fully understood, but arguably the most likely hypothesis is based on the radical pair mechanism (RPM). Whilst the theoretical framework of the RPM is well-established, most experimental investigations have been conducted at fields several orders of magnitude stronger than the Earth's. Here we use transient absorption spectroscopy to demonstrate a pronounced orientation-dependence of the magnetic field response of a molecular triad system in the field region relevant to avian magnetoreception. The chemical compass response exhibits the properties of an inclination compass as found in migratory birds. The results underline the feasibility of a radical pair based avian compass and also provide further guidelines for the design and operation of exploitable chemical compass systems.

[1] Centre for Advanced Electron Spin Resonance (CÆSR), Department of Chemistry, University of Oxford, South Parks Road, Oxford OX1 3QR, UK. [2] School of Molecular Sciences, Department of Chemistry and Biochemistry, Arizona State University, Tempe, AZ 85281, USA. [3] Physical and Theoretical Chemistry Laboratory, Department of Chemistry, University of Oxford, South Parks Road, Oxford OX1 3QZ, UK. [4] These authors contributed equally: Christian Kerpal, Sabine Richert. Correspondence and requests for materials should be addressed to C.R.T. (email: christiane.timmel@chem.ox.ac.uk)

In recent years, the field of animal magnetonavigation has been the focus of lively interdisciplinary research involving zoologists, chemists and physicists. Despite this, a comprehensive picture of the complex phenomena driving the ability of birds and many other animals to exploit the Earth's magnetic field for orientational and navigational purposes is yet to emerge. Two main hypotheses, one based on magnetite[1–7] and the other on photo-initiated quantum processes[8–11], are presently the front runners in this debate.

Originally proposed by Schulten in 1978[8], the avian quantum compass is based on the magnetic field-dependent quantum dynamics of short-lived radical pair intermediates, as governed by the so-called radical pair mechanism (RPM)[12,13]. Schulten's original proposal, first considered as exotic if not implausible by many, became of significant interest to the community only in the new millennium, following the discovery of cryptochrome, a blue-light photoreceptor protein[14], speculated to be fit for purpose as a chemical compass system[9].

A plethora of experimental and theoretical studies have now provided substantial support for the cryptochrome-based radical pair hypothesis[15]:

- Cryptochromes are expressed when birds perform magnetic orientation[16,17]
- The birds' ability to use the Earth's magnetic field is dependent on the wavelength of ambient light[18–22]
- Cryptochromes have been found in birds' retinae and studies on migratory birds' brains show that bilateral lesions of cluster N, a light-processing forebrain region, disables magnetic orientation in European robins[16,23,24]
- Weak radiofrequency magnetic fields can disrupt avian magnetic orientation[25–28] in agreement with the predicted results of the diagnostic tool for the action of the radical pair mechanism[26]
- Radical pair intermediates were observed in a number of proteins from the cryptochrome–photolyase family using time-resolved electron paramagnetic resonance[29–32]
- A radical pair reaction in a model system has been shown to respond to magnetic fields as weak as that of the Earth[33]

In addition, a variety of theoretical studies have provided a solid framework for the physical origin of the mechanism[9,34,35]. Yet, a number of crucial pieces of the puzzle are missing in the cryptochrome story, the most pressing of which probably regards the identities of the magnetic signalling state and cascade in any cryptochrome magnetoreceptor[15]. Similarly frustrating though, as concerning the very heart of the actual hypothesis, is the aforementioned absence of any proof of principle that the direction of a magnetic field can affect the yield and/or kinetics of a radical pair reaction in the so-called low-field regime in which the avian compass operates. As the RPM is the appropriate framework for the discussion of both high- and low-field regimes, it will be introduced below.

The suggestion that magnetic fields as weak as that of the Earth (30–65 µT) can affect certain chemical reactions, seems, at first sight, implausible, as the interaction of the Earth's magnetic field with a single molecule and at physiological conditions is orders of magnitude smaller than its thermal energy, $k_BT$. However, as we will see below, the reaction partners considered here are created in a highly polarised state far from thermal equilibrium, making such considerations irrelevant for the activationless processes to be described.

The RPM is concerned with the creation, field-sensitive evolution and reactions of a pair of radicals, created, most commonly, either by photoinduced homolytic bond cleavage or, as is the case for the system discussed below and in cryptochromes, by photo-induced electron transfer. Radical pair formation proceeds under conservation of total spin angular momentum so that a singlet (triplet) molecular precursor results in the formation of radicals with antiparallel (parallel) spins. The initially formed radical pair (for all further discussion from now on assumed a singlet) is created in a highly spin-polarised, non-eigenstate of the Hamiltonian and consequently begins to evolve coherently between singlet and triplet states, at a rate determined by the interactions between the electron spin and the magnetic nuclei within the radical pair (the hyperfine couplings). The application of an external magnetic field perturbs the efficiency of this singlet–triplet interconversion.

If the fields applied are weaker than the radical pair's hyperfine couplings (subsequently referred to as low or weak fields), symmetry breaking lifts the degeneracies among some of the zero field eigenstates and increases the number of pathways for singlet to triplet interconversion[36], giving rise to the so-called low-field effect (LFE)[35,37,38].

Conversely, in fields exceeding the radical pair's hyperfine couplings (subsequently referred to as high or strong fields), the singlet state becomes energetically isolated from two of the three triplet states (Zeeman effect), suppressing efficient singlet–triplet mixing.

Crucially, the effects of a magnetic field on the singlet–triplet mixing in both field regimes can only be observed under a number of strict conditions: (i) singlet and triplet radical pair states have different fates, either in terms of their recombination rates back to the ground state or their actual recombination products; note that only singlet radical pairs have the correct spin orientation to recombine to the ground state directly; (ii) relaxation (loss of spin coherence in particular) has to be slow on the timescale of both singlet–triplet mixing and radical recombination/reaction; (iii) radical recombination/reaction has to be slow on the timescale of the coherent spin evolution; (iv) the interaction between the radicals has to be weaker than their interaction with both the field and the magnetic nuclei.

While the effects of strong magnetic fields have been investigated and understood in depth, we are only now making progress to obtain a more complete picture of the (typically much less pronounced) sensitivity to weak fields. Experimentally, only one study on a model system has succeeded in providing proof for an (isotropic) Earth strength effect, while an orientation dependence of the magnetic field effect (MFE) was only observed for fields >3 mT[33]. The orientation dependence in this high-field region is caused by anisotropic hyperfine couplings in the radical pair, the anisotropic dipolar coupling being negligible compared to hyperfine couplings or indeed their anisotropies ($D \approx 0.06$ mT for the centre-to-centre distance of 3.6 nm in this pair)[39]. Previously only founded in theoretical simulations, it is speculated that these anisotropic hyperfine couplings in a radical pair with restricted motion may result in an orientation-dependent magnetic field response even in extremely weak magnetic fields including that of the Earth[9].

Using a custom-designed transient absorption (TA) spectrometer, we verify this hypothesis by testing if a quantum compass can function in fields as weak as that of the Earth. This is not only crucial regarding the discussion of the magnitude of any expected effects, but, importantly, the quantum dynamics in high- and low-field regimes are dominated by different processes[36]. The previously demonstrated existence of a chemical compass response of certain radical pair-based reactions in high fields[33] is therefore a necessary condition but by no means sufficient to explain the avian compass sense within the quantum system's low-field regime.

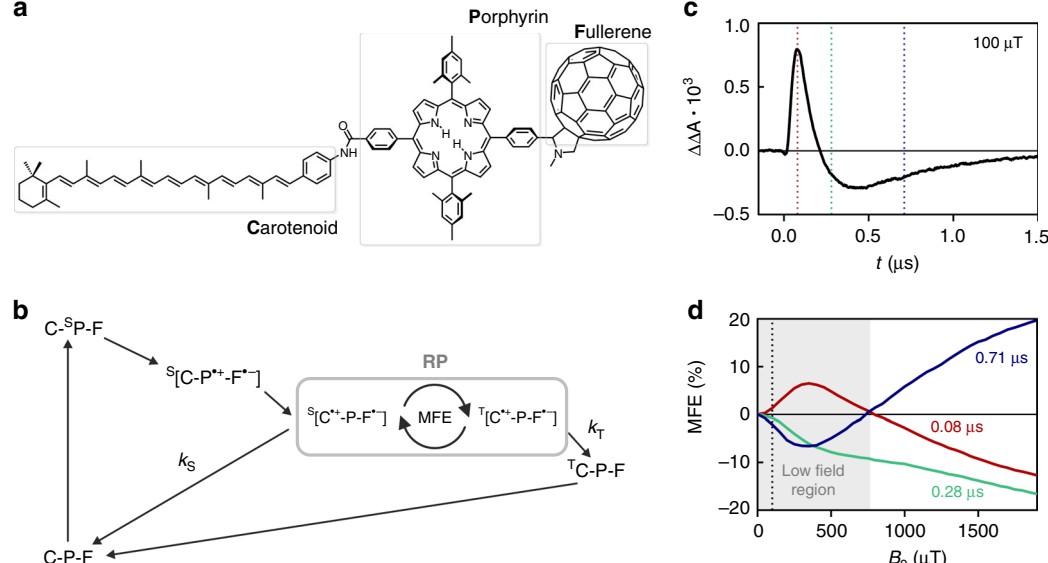

**Fig. 1** Chemical structure, photocycle, and time dependence of the magnetic field effect (MFE) of **CPF**. **a** Structure of the molecular **CPF** triad. **b** Simplified photoscheme including all processes of relevance for this study. For simplicity, the secondary radical pair $C^{\bullet+}$-P-$F^{\bullet-}$ is shown as created in a pure singlet state. For more detailed photochemical information, refer to refs. [40,41]. **c** Transient absorption subtraction signal $\Delta\Delta A$ of the radical pair, probed at 980 nm, obtained upon application of a magnetic field of $B_0 = 100\,\mu T$. The dotted lines in red, green, and blue indicate delay times of 0.08, 0.28, and 0.71 μs after laser excitation, respectively. **d** Magnetic field dependence of the MFE averaged for a time window of 20 ns centred around the indicated delay times. The dotted black line indicates the field position of 100 μT and the field region approximately corresponding to the low field region is highlighted in grey

## Results

**Chemical system and experimental conditions.** Figure 1a shows the structure of the investigated model chemical compass, a molecular triad consisting of covalently linked carotenoid (**C**), porphyrin (**P**) and fullerene (**F**) moieties. Its photophysical behaviour and response to high fields, in the absence and presence of resonant radiofrequency fields, have been studied previously[33,40–42]. As depicted in Fig. 1b, photo-excitation of the porphyrin at 532 nm is followed by rapid intramolecular electron transfer, first generating a primary radical pair C-$P^{\bullet+}$-$F^{\bullet-}$ of picosecond lifetime, before subsequent electron transfer leads to the formation of the secondary radical pair $C^{\bullet+}$-P-$F^{\bullet-}$, which lives for up to roughly a microsecond . Previous work, in similar solvent and temperature conditions to those employed here, demonstrated that this secondary radical pair is formed predominantly in the singlet state, $^S[C^{\bullet+}$-P-$F^{\bullet-}]$, with just 7% of radical pairs being created in the triplet state[41]. While each radical pair is born in a spin-correlated state (either singlet or triplet), the magnetic field characteristics of the radical pair ensemble are complex, as will be shown below.

The measurements were carried out at 120 K, where the solvent, 2-methyltetrahydrofuran (MTHF), forms an optically transparent glass. Recombination of $C^{\bullet+}$-P-$F^{\bullet-}$ is possible from either the singlet or triplet states and occurs with rate constants $k_S$ and $k_T$, respectively. The rates are strongly dependent on the solvent properties, notably its dielectric constant. Under similar conditions, $k_S$ has been shown to be some three orders of magnitude faster than $k_T$, and consequently, a significant change in the recombination kinetics is observed upon application of a magnetic field[41]. Further details on the sample preparation and experimental set-up are given in the Methods section.

Most experimental investigations of MFEs have relied on optical methods in which either the concentration of the radicals themselves or of one of their recombination products is determined as a function of field. Here we use nanosecond TA spectroscopy to obtain the concentration profile of the carotenoid radical cation $C^{\bullet+}$ via its absorbance in the near infrared,

following radical pair creation by a 532 nm laser pulse. Application of a magnetic field, $B_0$, is expected to change the rate of singlet–triplet interconversion and consequently the overall radical pair kinetics, concentration and absorbance. The effect of the field is typically quantified via $\Delta\Delta A(t, B_0) = \Delta A(t, B_0) - \Delta A(t, B_0 = 0)$, where $\Delta A(t, B_0)$ and $\Delta A(t, B_0 = 0)$ refer to the absorbance of the transient species at 980 nm (predominantly $C^{\bullet+}$), in the presence and absence of the field, respectively. $t$ defines the time after the 532 nm pump laser pulse. It can also be instructive to calculate the so-called percentage field effect, defined as $\text{MFE}(t, B_0) = \Delta\Delta A(t, B_0)/\Delta A(t, B_0 = 0) \times 100\%$.

**Time and field strength dependence.** Figure 1c demonstrates that a 100 μT field confers a pronounced effect on the recombination kinetics of the radical pair. In the presence of the magnetic field, the concentration of radicals immediately following the laser pulse is enhanced, $\Delta\Delta A(t < 220\,\text{ns}, 100\,\mu T) > 0$, but fewer radicals survive to microsecond timescales, $\Delta\Delta A(t > 220\,\text{ns}, 100\,\mu T) < 0$. This biphasic behaviour has been noted before and seems to be characteristic of singlet-born radical pairs with $k_S > k_T$ undergoing spin–lattice relaxation at a rate comparable to recombination[33,43]. The mixed initial spin state (93% singlet: 7% triplet) in $C^{\bullet+}$-P-$F^{\bullet-}$ further enhances this effect. Moreover, semiclassical spin dynamics simulations have recently reproduced some of the complex field- and time-dependent TA characteristics of $C^{\bullet+}$-P-$F^{\bullet-}$ without implicit consideration of relaxation processes or mixed initial spin states[36].

The percentage field effects $\text{MFE}(t, B_0)$, obtained at different times $t$ after laser excitation, are shown in Fig. 1d. Our initial discussion will concentrate on the data obtained at early and late times, i.e., $t = 0.08\,\mu s$ and $t = 0.71\,\mu s$. Both MFE traces follow the expected behaviour, with application of both low and high fields effecting changes in radical concentration of opposite sign. Following the discussion in reference[36], weak magnetic fields mainly enhance the $S–T_0$ interconversion efficiency, which results in an increase in radical concentration at early times after laser excitation (the initially predominantly singlet population is driven

more effectively into less reactive triplet) and in a decrease in radical concentration at late times (when formed triplet radical pairs can return more efficiently to singlet radical pairs, which might subsequently recombine).

In contrast, higher fields affect the radical recombination via the Zeeman effect, energetically isolating the $S/T_0$ manifold from the $T_+/T_-$ levels, therefore impeding efficient singlet–triplet mixing. Following the arguments above, this results in a decrease in radical concentration at early times after the laser pulse and a corresponding increase on longer timescales.

It is, at first sight, perhaps surprising that the MFE data obtained at intermediate times, namely, 0.28 μs after the laser pulse, do not exhibit a sign inversion of the MFE. This finding is, however, in agreement with the results in ref. [36] in which it was demonstrated that the field effects on the populations of $T_0$ and $T_+/T_-$ are not only in opposite directions but evolve at different timescales. While the initially positive LFE has, at 0.28 μs, already changed sign, the high-field effect lags about 0.2 μs behind in its evolution (see Supplementary Fig. 1).

Complementary experiments were also performed on a partially deuterated analogue of the **CPF** triad, from now on referred to as **CPF$_D$**, the chemical structure of which is shown in Supplementary Fig. 2. Significantly different MFE traces were recorded, while the kinetics of both triad molecules in the absence of any applied field were identical within experimental error (see Supplementary Fig. 3). The observed dependencies of the position and amplitude of the low- and high-field effects are in agreement with the reduced effective hyperfine coupling in the carotenoid radical (the gyromagnetic ratio of the deuteron is approximately a factor of 6.5 smaller than that of the proton). A reduction in this effective coupling is expected to result in a reduction of the size of the LFE and a shift of the maximum of the LFE to smaller fields[44,45], in agreement with Fig. 2b. A detailed interpretation of the difference in the time evolution of the field effects is more complex since the probabilities of finding the radical pair in the triplet and singlet states are determined by an intricate interplay of hyperfine couplings, applied magnetic field strengths and

kinetics[36]. For reference, the measured time-evolution of the MFE is compared for both triads in Supplementary Fig. 1 and a visualisation of the calculated proton hyperfine coupling tensors in the carotenoid radical is shown in Supplementary Fig. 4.

**The compass response.** We have so far characterised the triad's response to magnetic fields as a function of time and field strength. Crucial for chemical compass behaviour, we now investigate the dependence of the MFE on the orientation of the triad molecule with respect to the external field. The optically transparent glass formed by MTHF at 120 K immobilises the triad molecules on the timescale of the experiment. This allows the use of photo-selection to pump and probe only a subset of triad molecules with transition dipole moments predominantly parallel to the laser polarisation[33]. Previous calculations have shown that the transition dipole moments of the 532 nm porphyrin excitation and the $C^{\bullet+}$ absorption at 980 nm are almost parallel to the long axis of **CPF**, so that molecules aligned with the polarisation axis are preferentially excited and detected[33].

Orientation-dependent field effects are obtained as illustrated in Fig. 2a: (i) Parallel, linearly polarised pump (532 nm, green) and probe (980 nm, red) beams along the laboratory $y$ direction excite the sample and detect the field-dependent absorption of $C^{\bullet+}$, respectively. (ii) The applied magnetic field, at constant magnitude but varying orientation, is generated by three mutually perpendicular pairs of Helmholtz coils; the field strength along the laser beam path is 0 mT, and the field orientation in the $xz$ plane, perpendicular to the propagation direction of the laser beams, can be varied at will. Crucially, the magnetic field strength was calibrated to within an accuracy of $<\pm10$ μT at all orientations. (iii) The MFE was recorded as a function of the angle $\theta$ (in random order) between the field and the $-z$ axis of the laboratory frame.

The results of such measurements on the protonated triad (red graph) and deuterated triad (orange graph) are shown in Fig. 2c–e. All curves are shown for early times (0.08 μs after

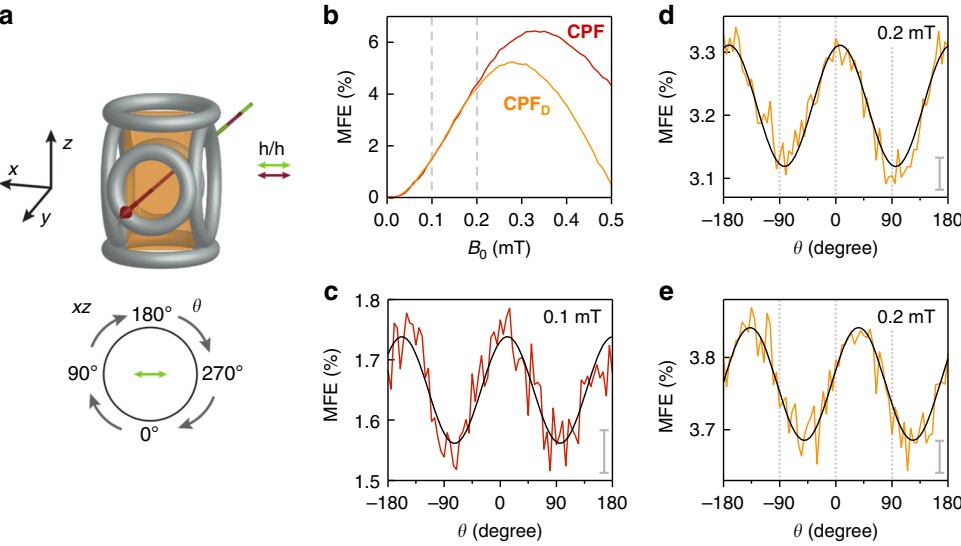

**Fig. 2** Orientation dependence of the magnetic field effect (MFE) of **CPF**. **a** Illustration of the experimental geometry. $x$, $y$ and $z$ indicate the three laboratory axes, $\theta$ defines the angle between the $-z$ axis and the applied magnetic field. The polarisation axes of the pump and probe beams are parallel and are indicated in the figure. **b** MFE for **CPF** (red) and **CPF$_D$** (orange) as a function of applied field. The field positions where the anisotropy curves were recorded are indicated by vertical dashed lines. **c** MFE of **CPF**, recorded at 100 μT as a function of $\theta$. **d** MFE data for **CPF$_D$**, recorded at 200 μT as a function of $\theta$. **e** MFE recorded for **CPF$_D$** and same parameters as **d** but recorded with the polarisation directions of both pump and probe beam rotated by 45° in the $xz$ plane. The data shown in all panels were averaged over a time window from 0.07 to 0.09 μs after laser excitation. The grey bars in panels **c**–**e** indicate the (average) standard deviation of the mean. The black solid lines represent best sinusoidal fits to the data

laser excitation). Unless otherwise stated, the data have been recorded with the pump and probe beams polarised along the $x$ axis of the laboratory frame, corresponding to a horizontal polarisation ($h/h$ in Fig. 2a).

For reference, the graphs in Fig. 2b indicate the field responses of the triads in the low-field region recorded 0.08 µs after the laser pulse. These data illustrate that the LFE is shifted towards smaller fields in the deuterated triad, as expected given its reduced hyperfine couplings.

The magnetic field responses of triads **CPF** and **CPF$_D$** with respect to angle $\theta$ are shown in Fig. 2c and d, respectively. The data were recorded using field strengths of only 100 µT for **CPF** and 200 µT for **CPF$_D$**. The respective data recorded at 50 µT are shown in Supplementary Fig. 5.

It can clearly be seen from the data that both triads act as sensitive weak-field chemical compasses: The MFE is expected[33] to be proportional to $\sin^2\theta$, with $\theta$ being defined in Fig. 2. This can be expressed as a simple sinusoidal variation of the MFE, $\sin^2\theta = 0.5 - 0.5\sin(2\theta + \pi/2)$, justifying the sinusoidal fit to the experimental data shown in black in Fig. 2. The MFE response of both triad systems shows a pronounced dependence on the angle $\theta$. Both chemical compasses are invariant to inversion of the field direction so that $\text{MFE}(\theta) = \text{MFE}(\theta + \pi)$, a characteristic shared by the avian magnetic compass[33,46].

Finally, Fig. 2e illustrates, on the example of **CPF$_D$**, that the observed angular dependence of the MFE response is driven by anisotropic hyperfine couplings. The data were recorded after a 45 degree (clockwise) rotation of the polarisation direction of the light within the $xz$ laboratory plane (with respect to $h/h$ polarisation). All other experimental parameters (including the orientation of the field coils) were kept the same. The corresponding 45 degree shift in the resulting compass response confirms the robustness of our calibration and affirms the chemical compass properties of the triad system.

## Discussion

The present study provides the first unequivocal experimental proof of a radical pair-driven compass operational at magnetic field strengths relevant to avian magnetoreception. The orientation dependence of the hyperfine-driven MFE response is demonstrated in the low-field region at fields comparable to that of the Earth on two model triad systems and for two different field strengths. The robustness of the calibration of the small fields in both magnitude and direction is ascertained by test experiments.

Having demonstrated the feasibility of a chemical compass response at such weak fields in a model system, the uncertainty regarding the existence of such anisotropic effects in a natural magnetosensor in ambient field conditions remains. The most likely avian magnetosensor is arguably the flavin/tryptophan radical pair generated by blue light excitation in the protein cryptochrome[10,47–51]. Yet field effects in vitro, in the isolated, intact proteins, have been only observed in isotropic solutions and at fields exceeding that of the Earth by a factor of approximately 20, with only one study providing experimental proof for a low-field feature[52]. The suggestion of a DNA-based magnetic sensor[53], albeit remarkable, poses significant questions, such as a surprisingly short lifetime of the radical pair and small distance (large exchange interaction) between the radical partners, both directly at odds with the requirements for a (low) field-sensitive radical pair[54].

The model triads studied here most certainly derive their exceptional compass sensitivity from a number of beneficial properties, namely: (1) Spin relaxation and radical pair kinetics that are slow enough to allow for efficient spin evolution, (2)

strong axiality of the dominant hyperfine coupling(s) in one of the radical pair partners (here **C$^{\bullet+}$**), and (3) absence of any significant hyperfine couplings in the other (here **F$^{\bullet-}$**). Clearly, these properties and our results presented here can serve as useful guidelines for designing and analysing other highly sensitive artificial chemical compass systems.

In the context of animal magnetoreception, one can well imagine that relaxation and recombination rates for the cryptochrome radical pair might have been optimised for function by evolution, e.g., through interaction with binding partners, slight variations of protein structure as well as solution accessibility (protonation/deprotonation) of the radicals (especially the terminal tryptophan)[55]. The second condition of strong axiality of the hyperfine couplings is, at least to some degree, fulfilled by the flavin radical[56]. However, a hyperfine-coupling-free second radical is harder to imagine. The frequently evoked hypothesis of the involvement of a superoxide radical, $O_2^{\bullet-}$, fails, as demonstrated in the literature[57], owing to the large spin–orbit coupling in $O_2^{\bullet-}$ leading to extremely fast electron spin relaxation. As a result, all spin coherence is lost on a nanosecond timescale and with it any magnetic field sensitivity.

One further hypothesis regards the involvement of the ascorbyl radical characterised by few and small isotropic hyperfine couplings[58]. In solution, the flavin/ascorbyl pair demonstrated sensitivity to weak fields much exceeding previously reported effects in other flavin-containing radical pairs, including cryptochromes. However, recent molecular dynamics simulations suggest that the brief and infrequent encounters of the ascorbyl radical with cryptochrome make this also an unlikely candidate in the search for an Earth strength field sensor[59].

We are hence at crossroads: experimentally we have proven (i) the sensitivity of light-induced radical pairs in cryptochromes to applied magnetic fields[52,60,61], if an order of magnitude larger than that of the Earth, and (ii) with this study, we have demonstrated, for the first time, the feasibility of a radical pair compass functional in fields relevant to magnetoreception, if in a model system. Any further studies in the discussion of a radical pair-based magnetoreceptor in cryptochrome need to be aimed towards high sensitivity measurements of such weak-field compass responses in oriented or otherwise immobilised proteins. Importantly, such techniques need to provide information on the identity of the radical pair partners involved and will most likely demand the application of optical or magnetic resonance techniques, which allow the identification of the involved radicals via their spectral fingerprints. These studies might well not be successful on the isolated proteins in vitro but demand the investigation of these systems in cellular environments where binding partners and solution conditions match those found in vivo.

## Methods

**Sample preparation**. The **CPF** triad molecule was synthesised as previously reported[40]. The MFE experiments detected by TA were performed at 120 K in MTHF. The solvent was purchased inhibitor free (Sigma-Aldrich, anhydrous, ≥99%) and kept under argon atmosphere. Any formed peroxides and traces of water were removed by passing the solvent over a column of activated alumina prior to use. A solution of **CPF** in MTHF was prepared at an optical density of about 0.4 at the excitation wavelength (3 mm optical path) and transferred into a custom-made rectangular optical cell with an optical path length of about 3 mm. Oxygen was removed by several freeze–thaw cycles and the cell was subsequently flame sealed under vacuum.

**Experimental set-up**. The samples were excited at 532 nm using a Nd:YAG laser operated at a repetition rate of 10 Hz (7 ns pulse duration), employing pump energies of <1 mJ. The radical pair kinetics were monitored via the absorption of the carotenoid radical cation at 980 nm using a modulated cw laser diode with a duty cycle of only 4% (to avoid excessive sample heating), corresponding to peak energies of ~3 mW at the sample during the measurement. The transmitted light was detected using an infrared-sensitive photodiode.

The magnetic field was generated using three sets of orthogonal Helmholtz coils. A small pair created a static field in the propagation direction of the pump and probe laser beams ($y$ direction) to cancel out the component of the Earth's magnetic field in this direction. Two larger sets were used to precisely control the magnetic field strength and direction applied in the experiment in the plane perpendicular to the laser propagation direction ($xz$ plane). A field angle $\theta$ of 0° corresponds to a field in $-z$ direction, whereas a field angle of 90° indicates a field applied in the $x$ direction. Details on the field calibration procedure are given in Supplementary Note 1.

The laser polarisations were controlled by making use of $\lambda/2$ waveplates. The pump and probe beam paths were collinear at the sample cell and decoupled after the sample prior to signal detection using dielectric mirrors (HR 532 nm). The used beam geometry maximised the pump–probe overlap at the sample. The polarisations of pump and probe beams were chosen to be parallel to simplify the analysis of the anisotropic data. A detailed scheme and description of the experimental set-up can be found in Supplementary Fig. 6 and Supplementary Note 2.

## Data availability

The data that support the findings of this study are available from the corresponding author C.R.T. on request and have also been deposited in https://www.ora.ox.ac.uk.

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

## Acknowledgements

We would like to thank D.E. Manolopoulos and T. Player for helpful discussions. C.K. is grateful for a DFG Research Fellowship (Project No. 256837888). The authors would like to thank the EPSRC (EPL011972/1), DARPA (QuBE: N66001-10-1-4061) and the US Air Force (USAF) Office of Scientific Research (Air Force Materiel Command, USAF Award No. FA9550-14-1-0095) for financial support.

## Author contributions

C.K. and S.R. built the experimental set-up, performed the experiments and analysed the data. J.G.S. developed the magnetic field calibration procedures. S.P., P.A.L. and D.G. synthesised the molecules. C.R.T. and P.J.H. designed the study. C.R.T. coordinated the study. S.R., C.K., P.J.H., S.R.M. and C.R.T. interpreted the data. C.R.T., S.R. and C.K. wrote the manuscript. All authors discussed the results and commented on the manuscript.

## Additional information

**Competing interests:** The authors declare no competing interests.

