## [Peer Review File · Nature Communications]

Reviewers' comments:

Reviewer #1 (Remarks to the Author):

The manuscript is an important extension of previous studies by Timmel, Hore and Gust of magnetic field effects in the charge separation and recombination in a donor-acceptor triad. The complex is a model for the postulated magnetosensor involved in avian navigation. Using polarized light to select specifically oriented molecules the authors demonstrate that the magnetic field effect depends on the orientation of the molecule relative to the magnetic field. Most importantly, they have demonstrated the orientation effect at weak magnetic fields comparable to the earth's field. Although there is plenty of indirect evidence to support the role of a radical pair based magnetosensor in birds, the orientation dependent magnetic field effect required for such a sensor to function has never been demonstrated experimentally in weak magnetic fields. Thus, the result represents a significant advance.

The paper is exceptionally well written and could be published essentially as is. However, I feel some additional details could be added to the supplementary information. Two very important factors for the success of the experiments are the control of the direction and strength of the magnetic field and the orientation selection by the polarized light. The authors say that they carefully calibrated the magnetic field but more detail of how this was done, and a quantitative measure of the field homogeneity would be useful. Similarly, the degree of polarization of the light should be reported.

I was also surprised that the orientation dependent data were collected at 0.1 mT and 0.2 mT, which are two and four times the earth's magnetic field strength, respectively. If the goal was to demonstrate that a magnetic field effect can be observed in the earth's field, why were the data not collected at 0.05 mT? It would be useful to know what the weakest field is at which the orientation dependence of the MFE could be detected.

Reviewer #2 (Remarks to the Author):

In the present contribution, Timmel and Hore demonstrate the sensitivity of a molecular magnetic compass molecule, a triad comprising linearly linked carotene, porphyrin and fullerene moieties, to applied weak magnetic fields comparable in strength to that of the Earth. Specifically the molecular model's orientation dependence towards the direction of the magnetic field is examined.

This is a logical continuation of previous experimental work published in 2008

in Nature and a theoretical treatment published last year in the Journal of Chemical Physics; a further excellent contribution from the Oxford labs.

The manuscript is beautifully written and deserves publication in Nature Communications as I feel it will be interesting to a wide readership.

Nevertheless, there are a few suggestions that the authors should consider:

(1) The analysis of the orientation dependence of magnetic field effects (MFE) and

the discussion of the results are rather qualitative. It is obvious that a deuterated carotene moiety exhibits weaker hyperfine couplings as compared to a protonated one.

Apparently, the anisotropy of the combined hyperfine interactions is crucial.

Could the authors be more specific than "... in agreement with the reduced effective

hyperfine coupling in the carotenoid radical" (page 6)? They could examine these couplings (at least theoretically) and predict the effect based on specific interactions.

It is (perhaps) surprising, that even the fully protonated molecule shows such strong MFE as I thought that numerous hyperfine interactions might quickly decohere singlet-triplet mixing. A few additional words of clarification might be helpful.

(2) Is there any (again more quantitative) information available on the angle segment

of molecules the polarised laser pump beam (532 nm) is able to excite? Statements such as "... with transition dipole moments predominantly parallel to the laser polarisation" (page 7, paragraph 1) and "... with the polarisation axis are preferentially excited and detected." (same page and paragraph) sound a bit vague.

(3) "The present study provides the first unequivocal experimental proof of a radical ***pair*** driven compass ..." (page 8, beginning of Discussion) is a quite strong statement, especially in light of the fact, that only the carotene radical is probed.

Perhaps an even more rigorous proof could be presented if an experiment were performed, in which a radio frequency field (suitable to drive transitions in the low-field region) is applied to "destroy" the MFE. Such an experiment would be similar to the one

conducted with migratory birds by Ritz and the Wiltschkos (Nature 2004).

(4) The statement "Finally, panel e) [of Figure 2] illustrates, on the example of CPF, that the observed angular dependence of the MFE is driven by anisotropic hyperfine couplings rather than being caused by errors in the field calibration." (page 7, last paragraph) needs a few more words of explanation, especially in light of the fact, that there are also other anisotropic interactions present in a radical pair, such as dipolar coupling. Furthermore, what are the errors in magnetic-field calibration?

(5) Referencing should be improved as citations are clearly biased towards

contributions from the own labs. There are several statements, particularly in the Discussion section, that lack references or that were referenced only to theoretical work by Hore and Timmel, e.g.:

On page 9, paragraph 3: "In the context ... of the radicals (especially the terminal tryptophan)." This statement needs several references, as a bunch of experimental data is available dealing with these aspects.

On page 2 and 3: It is surprising that the authors consider the occurrence of (long-lived) radicals in cryptochromes worth mentioning, but not that of radical pairs, which in the past 10 years have been detected in a number of cryptochromes, including the one from a magnetoreceptor.

On page 7, paragraph 5: "... a characteristic shared by the avian magnetic compass."

This statement needs (a) reference(s).

On page 8, last paragraph: "The most likely avian magnetosensor is arguably the flavin/tryptophan radical pair generated ... cryptochrome." This statement needs referencing.

Centre for Advanced Electron Spin Resonance

Professor Christiane Timmel
University of Oxford
OX1 3QR
Tel: + 44 1865 272682
christiane.timmel@chem.ox.ac.uk

Demonstration of a chemical compass in microtesla magnetic fields: a proof of principle for radical pair magnetoreception in birds

by

Christian Kerpel, Sabine Richert, Jonathan G. Storey, Smitha Pillai, Paul A. Liddell, Devens Gust, Stuart R. Mackenzie, Peter J. Hore, and Christiane R. Timmel

We are grateful for the positive feedback from both reviewers who believe that our paper “deserves publication in Nature Communications” and that it is “beautifully written” and “important”. We also believe that the comments and suggestions from both reviewers have helped us to improve our manuscript further. We show below how we have addressed all queries and suggestions raised by the reviewers:

Responses to reviewer 1:

1. [...] However, I feel some additional details could be added to the supplementary information. Two very important factors for the success of the experiments are the control of the direction and strength of the magnetic field and the orientation selection by the polarized light. The authors say that they carefully calibrated the magnetic field but more detail of how this was done, and a quantitative measure of the field homogeneity would be useful. Similarly, the degree of polarization of the light should be reported.

We thank the reviewer for her/his comment; the field calibration procedure indeed formed a major part of this project in terms of both time and effort. **We have now added a new section (Note 1) in the Supplementary Information explaining this procedure in detail.**

In brief: considering the dimensions of the coils and their arrangement (approximately Helmholtz) together with the small probe volume (beam diameter ~ 3 mm), field inhomogeneities are expected to be insignificant (a homogeneity in excess of 98% is estimated from the coil geometry of all three pairs of coils). In addition, during the calibration procedure, the magnetic field was measured at the position of the probe beam using a sensitive Hall probe with an active area coinciding with that of the probe beam.

The degree of polarisation of the light at the sample is essentially determined by the laser/laser diode output polarisation (close to 100 %). In order to maintain this high degree of polarisation, the use of reflective optics before the sample was kept to a minimum and the laser beam alignment was kept horizontal at all times. In order to test our own protocol we furthermore undertook the following test: The degree of linear polarisation is normally quantified by the so-called polarisation extinction ratio (i.e. the ratio of laser powers in the two polarisation directions measured by recording the power transmission of a polariser). When installing a polariser in the beam path before the sample and turning this polariser by 90° with respect to the laser

polarisation, we were able to reduce the power transmission to the noise level of our power meter. We therefore conclude that the degree of polarisation of the light is superior compared to the extinction ratio of the Glan-Taylor polariser used for the measurement ($10^5:1$).

2. I was also surprised that the orientation dependent data were collected at 0.1 mT and 0.2 mT, which are two and four times the earth's magnetic field strength, respectively. If the goal was to demonstrate that a magnetic field effect can be observed in the earth's field, why were the data not collected at 0.05 mT? It would be useful to know what the weakest field is at which the orientation dependence of the MFE could be detected.

We agree with the referee that measurement of an anisotropic field effect at 0.05 mT seems important and relevant as it corresponds to the "average" geomagnetic field (the latter varies from 0.025 mT to 0.065 mT across the planet). However, the radical pair system we study here is not a biologically relevant system and consequently the 0.05 mT geomagnetic field strength does not hold a specific relevance for this particular radical pair. Rather the detection of an anisotropic magnetic field effect in the so-called low field region is the objective of our study.

Nonetheless, we were also intrigued by this experiment and did indeed succeed in measuring the orientation dependence of the magnetic field effect (MFE) also at 0.05 mT. **We now include these data in the Supplementary Information.** It is important to realise that the MFE at 0.05 mT is less than a quarter of the MFE at 0.1 mT, significantly reducing the signal-to-noise ratio and pushing us to the very limits of our experimental capabilities. The data can still be reasonably fitted with a sinusoidal function and show maxima and minima consistent with the data for higher field strengths. However, unlike in all other (higher field) data, a slight asymmetry is observed in the orientational response for both CPF and CPF_D. The most likely cause for this (unphysical) asymmetry is that the magnetic field calibration reaches its limits here. For comparison, increasing the field strength from 0.05 to 0.06 mT in the magnetic field effect experiments changes the percentage MFE by about 0.2%, significantly exceeding the "asymmetry" observed here (approx. 0.02 %).

Responses to reviewer 2:

1. Apparently, the anisotropy of the combined hyperfine interactions is crucial. Could the authors be more specific than "... in agreement with the reduced effective hyperfine coupling in the carotenoid radical" (page 6)? They could examine these couplings (at least theoretically) and predict the effect based on specific interactions.
It is (perhaps) surprising, that even the fully protonated molecule shows such strong MFE as I thought that numerous hyperfine interactions might quickly decohere singlet-triplet mixing. A few additional words of clarification might be helpful.

We thank the referee for pointing out that some of our arguments seem to be unclear. We will answer the issues raised in turn:

Firstly, our comment in the paper "*... in agreement with the reduced effective hyperfine coupling in the carotenoid radical*" does not refer to the anisotropic field response but rather the characteristics of the low field effect (LFE). As previously shown, both experimentally and theoretically (see for instance, Stass et al. *Chem. Phys. Lett.* **1995**, 233, 444-450 and Rodgers et al. *J. Am. Chem. Soc.* **2007**, 129, 6746-6755), the largest low field effects are expected when one radical has large hyperfine couplings whilst the other has none (or few and small couplings). Whilst the fullerene radical fulfils

the role of the “hyperfine devoid” radical partner, deuteration of the carotenoid radical leads to reduction of its effective hyperfine coupling, thus reducing the overall size of the LFE, as demonstrated in Figure 2b. **We have now included an additional sentence to clarify our statement and included the publications cited above.**

However, as demonstrated in our recent theoretical and experimental work on isotropic field effects in the CPF triad (Lewis et al. *J. Chem. Phys.* **2018**, *149*, 034103), the LFE and MFE evolve at different rates leading to a complex temporal evolution of the field effect. We have discussed the origin and characteristics of this behaviour in detail in our previous work, and believe that, for the present manuscript, a repeated discussion on this would distract from the main message on the anisotropic response of the system in the low field region. **However, for completeness, we now modified Figure S1 in the Supplementary Information to show the full temporal evolution of both LFE and MFE for both triads.**

We agree with the reviewer that it would be useful if we “*could examine these couplings (at least theoretically) and predict the effect based on specific interactions.*” However, calculation of the anisotropic MFEs for a spin system of this size is simply not feasible. Approximately 50 nuclear spins contribute in the (protonated) carotenoid radical. Exact quantum calculations are therefore impossible. A semi-classical calculation could in principle be done but, given the size of the spin system, would still require very significant time and computing investment.

Secondly, we agree with the referee that *the anisotropy of the combined hyperfine interactions is crucial* for the anisotropic response. **The figure below (now also included in the Supplementary Information)** shows a visualisation of the proton hyperfine tensors in the CPF triad. The proton tensors shown in blue indicate the positions that are deuterated in CPF_D. As can be seen, the four protons marked with a red asterisk exhibit strongly symmetry related hyperfine anisotropies, responsible for the compass response. Deuteration of the triad (structure shown in the Supplementary Information) reduces a number of strong isotropic hyperfine interactions (as well as two of the anisotropic couplings). We had hoped (in agreement with the referee “*It is (perhaps) surprising, that even the fully protonated molecule shows such strong MFE*”) that deuteration of the methyl groups would eliminate a source of electron spin decoherence thereby increasing the MFE or indeed its orientation dependence. This is not observed suggesting that methyl group rotation is already fast enough at the temperature of the experiment (120 K) that it does not cause much spin relaxation.

2. Is there any (again more quantitative) information available on the angle segment of molecules the polarised laser pump beam (532 nm) is able to excite? Statements such as “... with transition dipole moments predominantly parallel to the laser polarisation” (page 7, paragraph 1) and “... with the polarisation axis are preferentially excited and detected.” (same page and paragraph) sound a bit vague.

The fully polarised laser beams each excite (or pump) with the usual $\cos^2\theta$ weighting where θ is the angle between the corresponding transition dipole moment and the electric field vector of the light. A full derivation of this and the resulting angular dependence of the singlet (or radical) yield are given in the SI of Maeda et al. *Nature* **2008**, *453*, 387-391. We **now refer to this paper in the main part of the manuscript for clarity.**

3. "The present study provides the first unequivocal experimental proof of a radical *****pair***** driven compass ..." (page 8, beginning of Discussion) is a quite strong statement, especially in light of the fact, that only the carotene radical is probed.

Perhaps an even more rigorous proof could be presented if an experiment were performed, in which a radio frequency field (suitable to drive transitions in the low-field region) is applied to "destroy" the MFE. Such an experiment would be similar to the one conducted with migratory birds by Ritz and the Wiltschkos (*Nature* 2004).

The referee refers to our suggestion of "A *diagnostic tool for the radical pair mechanism*" (Henbest et al. *J. Am. Chem. Soc.* **2004**, *126*, 8102-8103), a method proposing the simultaneous application of static and resonant radiofrequency fields when "*there is no prior knowledge of the nature and properties of the radical reaction*". This paper shows also that radical pair reactions "*exhibit a Zeeman resonance at a frequency that is not strongly dependent on the hyperfine interactions provided they are weaker than the applied static field*". At the static fields considered here, this condition is not fulfilled and for a spin system of the size of the triad it is not possible to predict the effect of a radiofrequency field on the radical (or radical yield) (Hiscock et al. *Biophys. J.* **2017**, *113*, 1475-1484), and the effect is most likely to be non-specific (*i.e.* no Zeeman resonance observed). However, we did demonstrate exactly such resonant effects employing resonant radiofrequency fields for the triad system in somewhat larger magnetic fields (exceeding 1 mT) in Maeda et al. *Phys. Chem. Chem. Phys.* **2015**, *17*, 3550-3559. **We now include a reference to this paper in our manuscript.**

Furthermore, the referee is correct in pointing out that we only monitor one of the radicals (the carotenoid). However, its concentration shows a biphasic field dependence (both MFE and LFE), providing unequivocal proof for the radical pair mechanism. Our conclusion regards the previously missing demonstration of a radical pair driven compass operational at magnetic field strengths relevant to avian magnetoreception, *i.e.* within the LFE region.

4. The statement "Finally, panel e) [of Figure 2] illustrates, on the example of CFPD, that the observed angular dependence of the MFE is driven by anisotropic hyperfine couplings rather than being caused by errors in the field calibration." (page 7, last paragraph) needs a few more words of explanation, especially in light of the fact, that there are also other anisotropic interactions present in a radical pair, such as dipolar coupling.

The referee is of course correct that any inter-radical dipolar interaction could also give rise to orientational effects.

However, the dipolar coupling in these systems is very small, $D \approx 0.06$ mT (from a point-dipole approximation using a radical-radical distance of 3.6 nm, see Di Valentin et al. *J. Chem. Inf. Model.* **2005**, *45*, 1580-1588) and its contribution to the field effect and the orientational dependence is negligible as compared to the effect of the hyperfine couplings or indeed their anisotropies, see Efimova et al. *Biophys. J.* **2008**, *94*, 1565-1574.

Furthermore, the pronounced effect of deuteration on the low field region, in which we demonstrate the compass response, proves that the field sensitive response of the triad is hyperfine driven.

We have added a comment on this in the main part of the manuscript and included the references mentioned above.

Furthermore, what are the errors in magnetic-field calibration?

We agree with the reviewer that the statement "... rather than being caused by errors in the field calibration" might have been confusing without further explanation. **We therefore removed this part of the sentence in the main text and include the necessary background information on the field calibration procedure in the Supplementary Information as a new Note.**

If the field was not well calibrated, changing the field direction (angle θ) could also have an impact on the field strength. Consequently, an angle scan could artificially yield a sinusoidal curve even in the absence of any anisotropic interactions. In principle our chosen procedure for field calibration precludes this from happening (**cf. the new Supplementary Note**). Nevertheless, we wanted to present an unequivocal proof for the robustness of our calibration: A change in the direction of the (linear) laser polarisation can only result in a shift of the anisotropy curve if anisotropic molecular interactions are present in the system. The fact that we observe such a shift, confirms the presence of a radical pair driven magnetic field response.

5. Referencing should be improved as citations are clearly biased towards contributions from the own labs. There are several statements, particularly in the Discussion section, that lack references or that were referenced only to theoretical work by Hore and Timmel, e.g.:

We now include 15 further references, as detailed above and below.

On page 9, paragraph 3: "In the context ... of the radicals (especially the terminal tryptophan)." This statement needs several references, as a bunch of experimental data is available dealing with these aspects.

To support this statement, we now include the following reference [Hiscock et al. *Proc. Natl. Acad. Sci. USA* **2016**, *113*, 4634-4639], which discusses in detail the different ways the performance of the compass could have been optimised by evolution.

On page 2 and 3: It is surprising that the authors consider the occurrence of (long-lived) radicals in cryptochromes worth mentioning, but not that of radical pairs, which in the past 10 years have been detected in a number of cryptochromes, including the one from a magnetoreceptor.

We thank the referee for pointing out this issue. He/she is of course entirely right. It is the lifetime of radical pairs (in which the process of coherent singlet triplet mixing is affected by applied magnetic fields) rather than the lifetime of radicals (such as in the garden warbler cryptochrome (cf. Liedvogel et al. *PLoS ONE* **2007**, *2*, e1106) in which the long radical lifetime precludes any magnetic field effects as spin relaxation destroys the necessary singlet triplet mixing). **We have now removed this sentence and instead state the importance of the detection of an antiphase transient EPR signature typical of spin correlated radical pairs in a number of members of the cryptochrome / photolyase family.**

We added the following four references:

1) Gindt et al. *Biochemistry* **1999**, *38*, 3857-3866. 2) Biskup et al. *Angew. Chem. Int. Ed.* **2009**, *48*, 404-407. 3) Nohr et al. *Biophys. J.* **2016**, *111*, 301-311. 4) Weber et al. *Proc. Natl. Acad. Sci. USA* **2002**, *99*, 1319-1322.

On page 7, paragraph 5: "... a characteristic shared by the avian magnetic compass." This statement needs (a) reference(s).

We thank the reviewer for pointing this out and added the following two references in support of this statement:

1) Wiltshcko et al. *Science* **1972**, *176*, 62-64. 2) Maeda et al. *Nature* **2008**, *453*, 387-391.

On page 8, last paragraph: "The most likely avian magnetosensor is arguably the flavin/tryptophan radical pair generated ... cryptochrome." This statement needs referencing.

The reviewer is of course right that this statement should be supported by references. We added the following six:

1) Dodson et al. *Trends Biochem. Sci.* **2013**, *38*, 435-446. 2) Rodgers et al. *Proc. Natl. Acad. Sci. USA* **2009**, *106*, 353-360. 3) Ritz et al. *Procedia Chem.* **2011**, *3*, 262-275. 4) Phillips et al. *J. R. Soc. Interface* **2010**, *7*, S241-S256. 5) Ritz et al. *J. R. Soc. Interface* **2010**, *7*, S135-S146. 6) Liedvogel et al. *J. R. Soc. Interface* **2010**, *7*, S147-S162.

We believe that we have addressed all the reviewers' comments and hope that you find our work suitable for publication.

Christiane R. Timmel

(corresponding author)

REVIEWERS' COMMENTS:

Reviewer #1 (Remarks to the Author):

The authors have addressed the few suggestions in my original review and the manuscript is now acceptable for publication in my view.

Reviewer #2 (Remarks to the Author):

The manuscript should now be accepted for publication in the present form.

Authors' reply

We thank all reviewers again for their comments and suggestions. Since the reviewers requested no further changes to our previously submitted manuscript, the scientific content of the manuscript has not been modified since the last revision.